



# Global river flow data developed from surface runoff based on the Curve Number method

Raghu Vamshi[1], Kathleen McDonough[2], Susan A. Csiszar[2], Ryan Heisler[3], Katherine E. Kapo[1], Amy M. Ritter[1], Ming Fan[2], Kathleen Stanton[3]

[1]Waterborne Environmental, Inc., Leesburg, VA, 20175, USA
[2]Procter and Gamble, Cincinnati, OH, 45201, USA
[3]American Cleaning Institute, Washington, DC, 20005, USA

*Correspondence to*: Amy M. Ritter (rittera@waterborne-env.com)

**Abstract.** The availability of detailed surface runoff and river flow data across large geographic areas is needed for several scientific applications, such as refined freshwater environmental risk assessments. Some limiting factors in developing detailed river flow datasets over large spatial scales have been paucity of detailed input spatial data and challenges in processing of these data. The well-established USDA Curve Number (CN) method was applied for spatially distributed hydrologic processing to estimate surface runoff. Publicly available global datasets for hydrologic soil groups, land cover, and precipitation were spatially processed by applying the CN equations to create a global mean annual surface runoff grid of 50 meters. Runoff was spatially combined with global hydrology of catchments and rivers from publicly available datasets to estimate daily mean annual flow (MAF) across the globe. Estimated daily MAF were compared with measured gauge flow at rivers in several countries which showed good correlation ($R^2$ of 0.76 - 0.98). These flow estimates can be used for diverse applications at local watersheds to larger regions across the globe. The two spatial data products of this project representing MAF at the global scale are publicly available for download at https://doi.org/10.6084/m9.figshare.22694146 (Heisler, et al., 2023).

## 1 Background and summary

The impetus for this research was identified while evaluating detailed river flow data necessary to conduct aquatic risk assessments (USEPA, 2021) for chemicals disposed of down the drain in regions and countries where such flow data do not exist. The iSTREEM® model (Kapo et al., 2016) (https://www.istreem.org/) is used to conduct aquatic exposure assessments, as part of environmental safety assessments in the United States (U.S.) and is based on detailed river hydrology and river flow data derived from the NHDPlus V2 dataset (USEPA, 2012). The mean annual average river flow data (McKay, et al., 2012) in NHDPlus V2 was estimated from surface runoff based on precipitation and temperature variation from 1900-2008 (McCabe and Wolock, 2011). However, to conduct exposure assessments in other countries, such detailed river flow datasets, or the data are not publicly available. Other similar down the drain models used for exposure assessments in Canada (Grill, et al., 2016; Ferrer and DeLeo, 2017), United Kingdom (Kilgallon, et al., 2017), China



(Hodges, et al., 2012; Grill, et al., 2018), and Asia (Wannaz, et al., 2018) have identified the need for refined river flow data to better estimate chemical exposure to the aquatic environment.

Process driven approaches to estimate surface runoff have been applied at lower resolution spatial scales, primarily due to limitations in the availability of detailed input spatial data and computing resources (Bierkens, 2015; Beck, et al., 2017). Improvements in technology and computing power over the past few years have made it possible to apply data intensive approaches to create datasets not just over smaller geographies, but at global scales. These novel developments have made it possible to practically implement heavily process driven approaches to estimate surface runoff and river flow at a global scale. Spatial data are relevant as more detailed scale brings about improved accuracy in flow estimation thereby bridging the gap toward more accurate predictions at refined spatial scales.

There are several surface runoff and river flow datasets available at a global scale, each offering different benefits. Many of the process-based surface runoff datasets (Arnell, 1999; Vörösmarty, et al., 2010) consider the aspects of geomorphology and climate, but are available at a coarse spatial resolution of approximately 50 km and need further processing before they can be adapted for river flow. The global freshwater model WaterGAP (Alcamo, et al., 2003; Döll, et al., 2003) estimates river flows based on water storage, withdrawals and consumptive uses on a global scale, but is available at a medium spatial resolution of 10 km. A regression method was developed (Barbarossa, et al., 2017) to calculate mean annual streamflow that overcomes some of the past challenges of spatial scale by providing flows at a detailed catchment scale, but the use of coefficients and assigning equal weightage to the catchment-level variables may not account for all the local factors that influence flow. The most recent FLO1K (Barbarossa, et al., 2018) dataset provides streamflow at a resolution of approximately 1 km for the globe. It takes a different data-driven approach by fitting an artificial neural network regression on flow observations from monitoring stations and estimating streamflow across the upstream hydrologic network by using flow observations in combination with covariates of upstream catchment physiography.

While these approaches each offer unique benefits, our research explores an alternate approach using the Curve Number (CN) method developed by the U.S. Department of Agriculture (USDA) (USDA, 1986) and publicly available global datasets (Ross, et al., 2018; ESA and Université Catholique de Louvain, 2010; Fick and Hijmans, 2017). These datasets offered an opportunity to combine the CN method towards creating a global surface runoff with detailed global hydrology datasets from HydroBASINS (Lehner and Grill, 2013) and HydroRIVERS (HydroRIVERS, 2019) to estimate river flow. Two advantages of this approach are that it does not rely on monitored flow data as an input and instead allows for evaluation of estimated flow accuracy using the monitored (measured gauge) flow data, and secondly, the use of detailed input spatial data captures the local variations of soils, land cover and precipitation thus providing an estimate of river flow reflective of local geographic conditions.

Presented here is a detailed description of the standardized framework, methodology, global input datasets, and spatial processing steps used to estimate global river flows, as well as several comparisons of the results to monitored gauge data for evaluation. The data generated in this project is publicly available and ready for use in a diverse set of applications, including



down the drain environmental exposure modelling, and offers flexibility for applying at various spatial scales across the globe.

## 2 Methods

### 2.1 Surface runoff and river flow

The first step in estimating the amount of flow in a stream or river is to quantify the amount of surface runoff feeding into
the stream or river which is influenced by several inter-related physical factors.  Water flows occur on land surface when there is more water than the land can absorb, and the water reaches the nearby creeks, streams, rivers, lakes, and other surface waterbodies.  Runoff varies temporally and spatially based on natural and anthropogenic factors including rainfall intensity and duration, soil characteristics, land cover, slope, and drainage area.  In the U.S., variability in surface runoff has been predominantly attributed to variation in rainfall (McCabe and Wolock, 2011). Runoff datasets have been used to derive
river flows in the U.S., the NHDPlus V2 dataset (USEPA, 2012) being a recent example of this; however, such detailed datasets are not available at the global scale and a methodology for estimating surface runoff and eventually river flow leveraging this unique combination of data was needed.

### 2.2 Curve number and data

The Curve Number (CN) method (USDA, 1986 and 1997) developed by the USDA Soil Conservation Service in the 1950s
provides a simplified approach to estimate key hydrologic processes while being grounded in a physical representation of saturated flow and runoff processes (Ponce and Hawkins, 1996; Garen and Moore, 2005). This method is based on algorithms for estimating surface runoff for a given unit of area and is a function of soil group, land cover complex, and antecedent moisture conditions from rainfall. The method was originally developed to determine the quantity of direct runoff in rural and urban watersheds from a specified amount of rainfall. The CN method is a well-established approach that has
been used for ungauged watersheds since the rainfall and watershed data are generally more readily available and easier to obtain (USDA, 1997).  The CN method avoids the problems inherent to parameterizing and running more complex models due to its simplicity and relatively low data input requirements, and has been implemented in a variety of hydrologic, erosion, and water quality models (Carousel, et al., 2005; Knisel and Davis, 1999; Arnold et al., 2012; Steglich, et al., 2019). This method of estimating rainfall excess from rainfall-runoff relationships is widely used in applied hydrology (Ponce, et
al., 1996; USDA, 1997; Burke, 1981).  Intricate variability in the rainfall-runoff relationship has been shown to be well represented by the CN method (Young and Carleton, 2006) which has led to its adoption into complex regulatory models including the USEPA model for pesticide risk assessment, Pesticide in Water Calculator (USEPA, 2020).

The CN method was selected as the optimal approach to achieve a runoff estimation that was scalable to accommodate the best available data resources across geographies with the finest resolution.  This method was adopted in this study because,



firstly, it has been widely used throughout the U.S. (Verma, et al., 2017; Woodward, et al., 2003; Tedela, et al., 2012) and several regions across the world (Hawkins, 1984, Brocca et al., 2009; Fan, et al., 2013; Lal et al., 2015).  Secondly, it has been tested on watersheds of varying sizes (Tedela, et al., 2012).  Finally, the input only requires readily available spatial datasets.  The last aspect was an important consideration on this project due to the need to expand this approach to the global scale and availability of detailed input spatial datasets at that scale were limited.

In short, the CN method uses two equations to estimate surface runoff ($Q_r$):

$$S = (1000/CN) - 10 \qquad\qquad (1)$$

where $S$ is potential maximum retention after runoff begins (inches) and $CN$ is the curve number value of the land cover type ranging from 0 to 100.

$$Q_r = (P - 0.2S)^2/(P + 0.8S) \qquad\qquad (2)$$

where $Q_r$ is runoff in inches, $P$ is rainfall in inches, and $S$ is the potential maximum retention after runoff begins in inches.

CN primarily considers the physical characteristics of a soil and land cover that occupies the landmass which directly influences the amount of rainfall that infiltrates into the soil and the remaining amount that aggregates at a receiving waterbody nearby as surface runoff.  The major factors that determine CN, and thus runoff, are soil classes, land cover type (e.g., cropland, forest, urban area, etc.), and precipitation, which are discussed in detail in the subsequent sections.  The

schematic in Fig. 1 provides an overview of the various datasets used in spatial processing, and Table 1 lists in detail the spatial datasets used for CN and flow estimation.  The CN-based runoff estimation method is not restricted for use in small watersheds, and it applies well to other large areas if the geographical variations of rainfall, soil and land cover complex are considered (Garen and Moore, 2005).  In using the CN method, we did not identify limitations on the size of the watershed, but the rainfall should be uniform across the watershed (Burke, 1981). Therefore, this is best accomplished by working with

smaller hydrologic units of a river basin which is discussed in the subsequent sections.




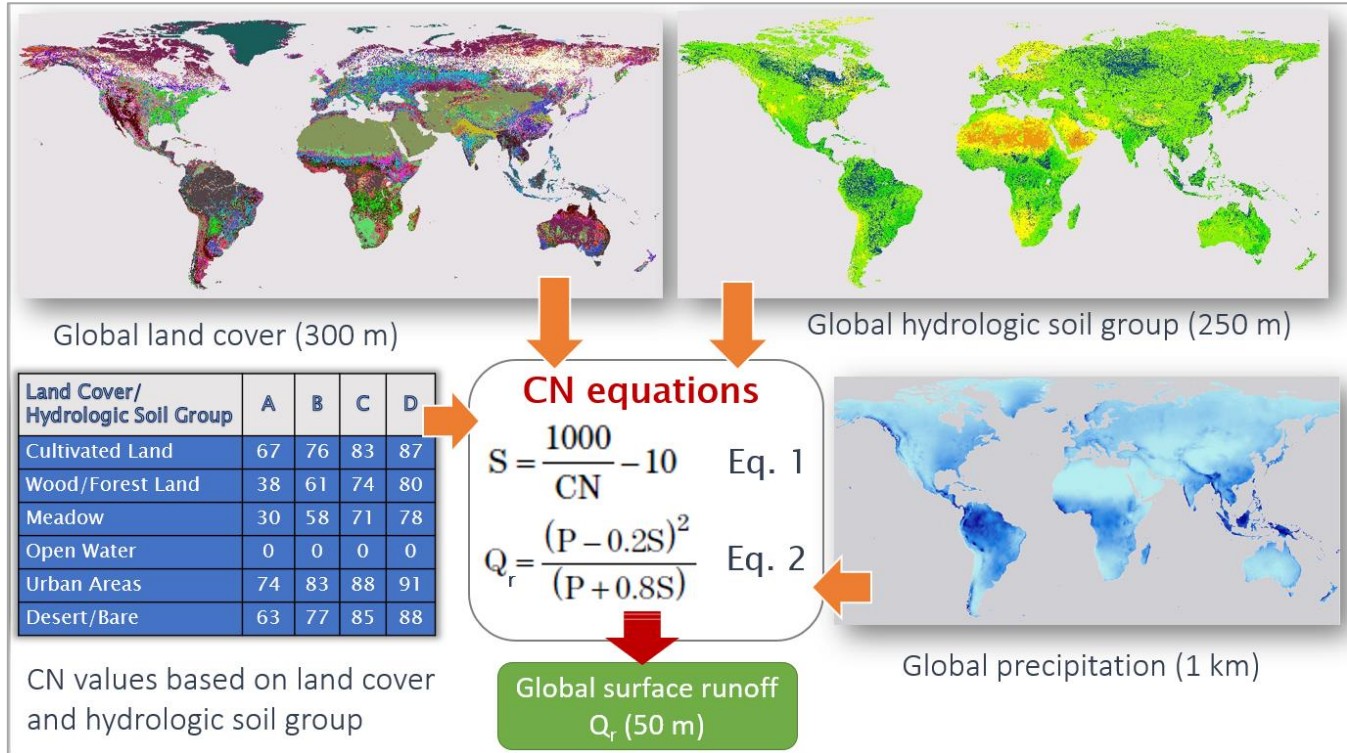

**Figure 1.** Overview of the CN approach to generate the 50 m runoff grid

**Table 1.** Datasets used in developing the CN mean annual surface runoff and river flow

| Attribute | Name | Resolution and Extent | Data Source | Description |
|---|---|---|---|---|
| Soil Hydrologic Group | HYSOGs250m | ~250m gridded, global | Soil hydrologic groups: HYSOGs 250m dataset (Ross et al., 2018) | Gridded soil hydrologic groups developed for curve number-based runoff estimation |
| Land Cover | GlobCover 2009 | ~300 m gridded, global | Land cover: ESA GlobCover 2009 land cover dataset (ESA and Université Catholique de Louvain, 2010) | Land cover classifications with 22 classes based on annual MERIS (Medium Resolution Imaging Spectrometer Instrument) fine resolution surface reflectance mosaics for 2009 |
| Precipitation | WorldClim 2 | ~1 km gridded, | Precipitation: WorldClim 1 | Mean monthly precipitation data |



| | | global | km precipitation dataset (Fick and Hijmans, 2017) | representing the time period from 1970-2000 |
|---|---|---|---|---|
| Hydrology | HydroBASINS and HydroRIVERS | Vector, global | Hydrology: HydroBASINS (Lehner and Grill, 2013) and HydroRIVERS (HydroRIVERS, 2019) datasets | Spatial hydrologic dataset for watershed boundaries and sub-basins, and river network features |

### 2.2.1 Soils

A key input needed to determine the CN in Eq. (1) is the Hydrologic Soil Group (HSG), which represents the minimum water infiltration rate of a given soil type. Surface runoff is affected by how much moisture a given soil type can retain, and

depends on soil properties such as permeability, texture, and compaction. HSGs are divided into four groups, A through D, with group A having the lowest runoff potential and D having the highest runoff potential (USDA, 1986; Ross, et al., 2018). Some countries have a publicly available soils database with HSGs listed such as the Soil SURveyGeOgraphic (SSURGO) database in the U.S. (Soil Survey Staff, 2009). However, most countries do not have readily available soils databases and the soil properties do not include HSG or the definition of HSG may not be consistent across countries. Therefore, the HSG

input used in this model was obtained from the HYSOGs250m (Ross, et al., 2018) raster HSG dataset which has their classification of HSG derived from soil texture classes at a spatial resolution of approximately 250 m and represent typical soil runoff potential at a global scale. According to Ross et al. (2018), the process for producing HYSOGs250m consisted of classifying HSGs from USDA-based soil texture classes and specifications, and took into account soil texture, depth to bedrock, and groundwater table depth. This global dataset available in a gridded format was used for hydrologic soil group

classification inputs needed for CN estimation.

### 2.2.2 Land cover

In addition to soils, the land cover complex (e.g., cropland, forest, urban area, etc.) covering the landmass plays a key role in the hydrologic water cycle by controlling infiltration and is a key input needed to determine CN. The type of land cover in combination with the soil type directly influence the amount of runoff from precipitation. For example, croplands with clay

soils have higher runoff potential compared to croplands with sandy soils where water infiltration is higher, which results in lower runoff potential. Similarly, impervious surfaces (e.g., paved parking lots, driveways, roads, etc.) in urban areas have low possibilities for infiltration and result in higher runoff potentials (USDA, 1986). The GlobCover (ESA and Université Catholique de Louvain, 2010; Bontemps, et al., 2011) initiative of the European Space Agency developed a service for the generation of global land cover maps, based on satellite imagery. GlobCover's 2009 release offers 22 land cover classes

following the United Nations classification system, including cropland, forest, grassland and artificial surfaces covering the





globe at a resolution of 300 m. These data are also highly desirable for this application as it has been validated by an international group of land cover experts with the highest accuracies occurring in Europe (Bontemps, et al., 2011). Due to its wide uses, global coverage, validation of the dataset, and spatial resolution, this gridded dataset was used to meet the land cover needs for CN estimates.

### 2.2.3 Precipitation

In addition to the incorporation of soil characteristics and land use factors to estimate CN, the distribution and intensity of precipitation determines the amount of water over a land area and thus has a direct impact on the volume of surface water runoff. The WorldClim 2 (Fick and Hijmans, 2017) dataset offers spatially-resolved monthly climate data for global land areas at a spatial resolution of approximately 1 km (30 arc-seconds). Among other climatic variables, the dataset provides average monthly precipitation over a temporal range of 30 years (1970-2000), using data from 34,542 weather stations across the globe. The dataset was created by interpolating between weather station data taking into account satellite data on variables such as elevation, cloud cover, land surface temperature, and distance to a coast. This dataset was validated and had global cross-validation correlations of 0.86 for precipitation (Fick and Hijmans, 2017); therefore WorldClim 2 was used to represent the precipitation needed to estimate runoff based on the CN method. While the temporal resolution of this dataset is at a monthly scale, this dataset was leveraged because it represented highest temporal resolution spatial global dataset available for use as model input, and the data were converted into daily precipitation rates (inches per day) for use in Eq. (2).

### 2.2.4 Hydrology

The estimated flows were connected to river networks using the HydroBASINS (Lehner and Grill, 2013) and HydroRIVERS (HydroRIVERS, 2019) datasets. HydroBASINS is a global dataset that provides spatially resolved watershed boundaries, and sub-basin and catchment-level delineations. It consists of hierarchically nested river basins and catchments based on the widely accepted Pfafstetter coding system levels (Verdin and Verdin, 1999). The Pfafstetter system defines nested basin to catchment levels numbered from one to twelve globally, where level-1 refers to the largest unit (i.e., river basin) and level-12 refer to the smallest unit (i.e., river catchment). Given the need for this project to develop detailed river flows, the smallest level size or most detailed catchment (i.e., level-12 polygons) from HydroBASINS were identified as the spatial unit of analysis. The level-12 catchments in HydroBASINS ranged in size from 0.03 km$^2$ to approximately 1,220 km$^2$ with an average of approximately 130 km$^2$. HydroRIVERS (2019) is a subset of the HydroBASINS dataset and offers a suite of vector line networks depicting streams and rivers at the global scale. There exists a spatial and tabular relationship between the river lines in HydroRIVERS and level-12 catchments from HydroBASINS data layers, thereby linking the streams and rivers to their respective level-12 catchments. Due to the large size, both datasets are distributed at a regional scale to include nine regions covering the globe: (1) Africa, (2) North American Arctic, (3) Central and South-East Asia, (4) Australia and Oceania, (5) Europe and Middle East, (6) Greenland, (7) North America and Caribbean, (8) South America,



and (9) Siberia. Hydrology for both datasets, including all nine regions, was used as model input for river basins, catchments, and flow connectivity.

## 2.3 Curve number refinement

This section describes how the previously described hydrologic soil group and land use data were used to derive CNs for use in Eq.1. The hydrologic soil groups from HYSOGs250m (Ross, et al., 2018) covering the globe consisted of eight soil groups – four standard soil groups (A, B, C, and D) and four dual soil groups (A/D, B/D, C/D, and D/D). Dual soil groups were consolidated with their corresponding standard groups (i.e., A/D with group A, B/D with group B, etc.) for simplification and efficient processing of CN calculations. The hydrologic soil group and the fraction of imperviousness

area based on the land cover are the most significant drivers in determining the CN. The 22 land cover classes from GlobCover (ESA and Université Catholique de Louvain, 2010) covering the globe were aggregated to six land cover classes for efficient processing of CN calculations. The aggregation was done by grouping similar land cover types that had similar CNs into one group (within the same soil type), thus prioritizing imperviousness for runoff estimation. Standard CNs provided by the USDA (1986) were used for the consolidated hydrologic soil groups and land cover classes to develop a

unique CN value for each hydrologic soil group (A, B, C, and D) and land cover class combination. For example, separate land cover classes from GlobCover representing forests including broadleaved evergreen forest, broadleaved deciduous forest, needle leaved evergreen forest, and mixed broadleaved and needle leaved forest were aggregated and classified as a single wood/forest land class. These land classes would have had relatively small variation in CNs and thus for the purpose of this study to aid in efficient processing, they were aggregated into a single wood/forest land use class. Similar land use

aggregations were performed for cropland, grassland, and urban land cover classes. All CNs are based on Antecedent Moisture Condition II for average conditions (USDA, 1997). These CNs were aggregated for the hydrologic soil group-land cover combination and refined for further use (Table 2).

**Table 2.** Curve numbers for the combination of land cover class and hydrologic soil group

| Land Cover Class | Land Cover Description | Hydrological Soil Group – CN Value | | | |
|---|---|---|---|---|---|
| | | A | B | C | D |
| 1 | Cultivated Land | 67 | 76 | 83 | 87 |
| 2 | Wood/Forest land | 38 | 61 | 74 | 80 |
| 3 | Meadow | 30 | 58 | 71 | 78 |
| 4 | Open Water | 0 | 0 | 0 | 0 |
| 5 | Urban-average | 74 | 83 | 88 | 91 |
| 6 | Desert/bare | 63 | 77 | 85 | 88 |






Lower runoff is expected in areas with a low CN value, for example, meadow land with a hydrologic soil group of 'A' has a CN of 30. Similarly, higher runoff is a factor of high CN value, for example, cultivated crop land with a hydrologic soil group of 'D' has a CN of 87. Group A soils (more sand and less clay) have very low runoff potential and group D soils (less

sand and more clay) have high runoff potential. On a regional scale, the area represented by each soil group is highly variable (Table 3). At a global scale, 4% of the area is represented by group A soils, 13% by group B, 67% by group C and 16% by group D soils; showing the higher runoff potential soils comprise a larger proportion by area.

**Table 3.** Area covered by each hydrologic soil group across the region and globe

| Region | Hydrological Soil Group – Percentage of Total Area | | | |
|---|---|---|---|---|
|  | A (%) | B (%) | C (%) | D (%) |
| Africa | 14 | 24 | 41 | 21 |
| North American Arctic | 0 | 9 | 87 | 4 |
| Central and South-East Asia | 0 | 4 | 87 | 9 |
| Australia and Oceania | 0 | 3 | 66 | 31 |
| Europe and Middle East | 5 | 26 | 68 | 1 |
| North America and Caribbean | 0 | 16 | 67 | 17 |
| South America | 0 | 4 | 65 | 31 |
| Siberia | 0 | 6 | 93 | 1 |
| Global extent (total of all regions) | 4 | 13 | 67 | 16 |


Recently, the GCN250 dataset (Jaafar, et al., 2019) with global gridded CNs became available and was used to compare to results from the methods and land use dataset used in this study. The GCN250 dataset used a different underlying land use dataset with more land use classes, and thus provided an appropriate dataset to evaluate the land use (and thereby CN) aggregations used in this study. To do this, the CNs for soil group C (which was found to be the most predominant soil

group across the globe, Table 3) from this study were compared to those in GCN250. For each of the six land use types in this study (Table 2), the CNs from GCN250 set of the relevant land use types were averaged and then compared to the aggregated CN in this study. For soil group C, the average CNs from the GCN250 dataset were 75, 74, 82, Not applicable/0, 88, and 92 for 'Cultivated land', 'Wood/forest land', 'Meadow', 'Open water', 'Urban', and 'Desert/bare' land types, respectively; and in the same order, from this study the CNs were 83, 74, 71, 0, 88, and 85. Additionally, the small variation

in CN for forest types in our aggregation was also found in the GCN250 dataset where the range in CN tree plant functional types (PFT) was relatively small, for example for soil type C, CN ranged from 70-77 across the tree PFTs. The results from



this comparison indicate good agreement and that the land use type aggregation used in our study were reasonable, appropriate, and fit-for-purpose.

## 2.4 Surface runoff dataset

The ArcGIS (Esri, 2019) suite of software was used for spatial data management and processing operations. The coordination system was standardized across all three input datasets to bring them from their native projection system to the World Cylindrical Equal Area (WCEA) (GIS Geography, 2020) in ArcGIS. This projection system was chosen since it reduces spatial distortion at the Equator and poles, to keep any possible shift in raster values to the minimum, and for spatial operations over large areas that require equal-area representation. Hydrologic soils groups, land cover, and precipitation

were individually reprojected (using the 'project raster' function in ArcGIS) to the WCEA projection system. The precipitation dataset required additional processing to convert the source monthly precipitation data to an annual mean daily precipitation value for each grid cell. The hydrologic soil groups, land cover and newly created annual mean daily precipitation gridded datasets were resampled by applying the nearest neighbor method (using the 'resample' function in ArcGIS) from their native resolution to 50 m as a common denominator to normalize for variation in spatial resolution

across the source datasets. The nearest neighbor resampling method causes the least amount of error in the resampled output (Esri, 2021; Brandsma and Können, 2006), and thus was preferred for use in our work. Resampling was additionally done to avoid generalizing values from any pixel values that cross each other. From this step on, until the CN runoff was estimated, all spatial processing was performed at 50 m, and the final runoff data was created at 50 m resolution.

       The hydrologic soils group and land cover datasets were spatially reclassified (using the 'reclass by table' function in

ArcGIS) so the source soil groups, and land cover classes are adjusted into the newly consolidated four hydrologic soil groups and six land cover classes as outlined in Table 2. Both the reclassified datasets were spatially combined (using the 'combine' function in ArcGIS) to apply the unique CN values to all raster cells representing the soil group-land cover matrix. At this point each raster cell in the resulting dataset has a CN value for each combination of soil group and land cover class. Substituting for CN in Eq. (1) where CN represents cell value in the raster dataset just created, potential

maximum retention (S) value for each cell was calculated (using the 'raster calculator' in ArcGIS). An additional step was included as a check for the runoff condition in Eq. (2): If $P<0.2S$, then $Q_r=0$. Substituting for potential maximum retention (S) and mean annual daily precipitation (P) in Eq. (2), daily runoff ($Q_r$) in inches per day was calculated (using the 'raster calculator' in ArcGIS) for each grid cell at 50m spatial resolution covering the globe. The units of $Q_r$ were next converted from inches to millimeters (using the 'raster calculator' in ArcGIS) for a daily mean annual runoff (MAR) (Fig. 2(a)) for

subsequent flow estimation.



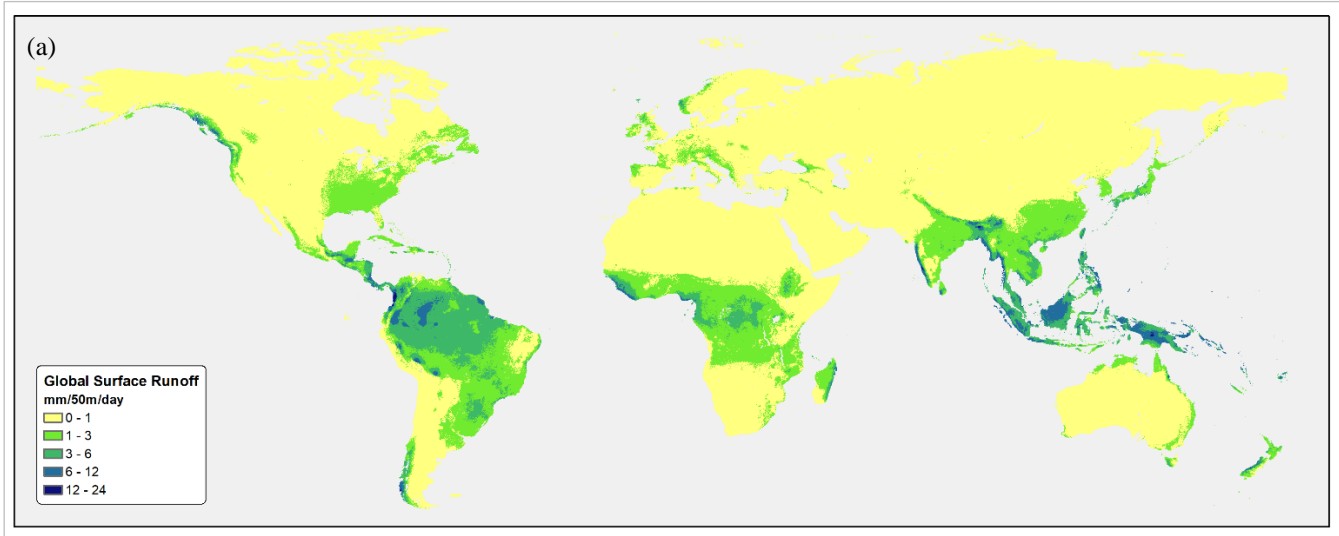

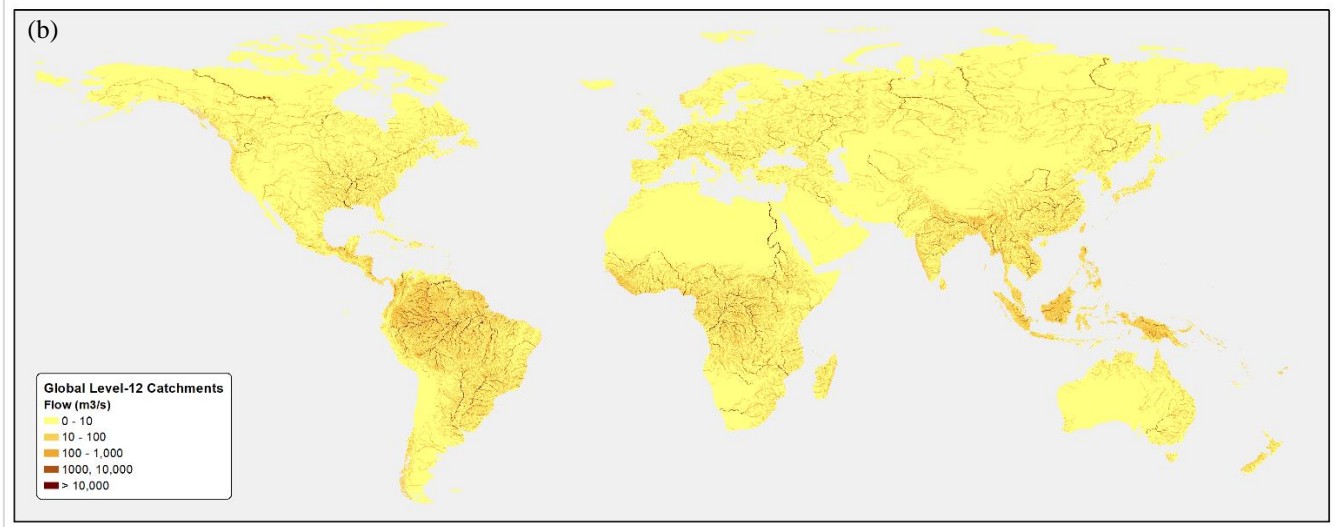

**Figure 2**. (a) Estimated global surface runoff grid (MAR, mm) at 50 m resolution based on the CN method. (b) Estimated Mean Annual Flow (MAF, m³s⁻¹) at HydroBASINS level-12 catchments covering the globe

The MAR dataset covers the global landmass and excludes parts of the Arctic and Antarctic regions due to non-availability of some of the input datasets for a particular geography. For example, hydrologic soils group data did not include Greenland, hence MAR was not calculated for that geography. Similarly, MAR was not estimated for some tropical islands including Hawaii and Mauritius due to non-availability of hydrologic soils group data for those areas. As to our knowledge, the MAR thus created is the highest spatial resolution (at 50 m) surface runoff dataset available at the global scale representing a combination of land cover and soils with the incorporation of precipitation over a 30-year temporal scale – accounting for the historic variability from dry and wet years to arrive at a best estimate of runoff under current conditions.





**2.5 Catchment MAF estimation**

With the MAR estimated for the globe, the next step was to estimate mean annual river flow. Flow in a stream or river at any location is an aggregation of the water runoff from a given surface area over a period of time (i.e., volume/time). At a large scale, the river basin determines the boundary to aggregate runoff and determines the amount of water represented as river flow discharging from the basin. At a small scale, hydrologic features including sub-basins and much smaller catchments comprise the larger river basin and provide the geographic variability of flow across the river basin. Using the

runoff grid, flow can be aggregated for individual catchments based on the spatial scale of interest, with smaller catchments providing more detailed flow at local scales, which was the need for this project. Hence, a global hydrologic dataset of catchments and rivers was required to convert the MAR gridded data (50 m) to river flow (i.e., volume flux). The level-12 catchment spatial vector polygons from HydroBASINS identified as the spatial unit of analysis were used for processing in the subsequent steps.

The level-12 catchments were spatially overlaid with the global MAR grid (using the 'zonal statistics as table' function in ArcGIS) to calculate individual catchment-level average value of the MAR (mm $d^{-1}$). This average MAR for each level-12 catchment was first converted to units of m$d^{-1}$ and then multiplied by the total geographic area of the level-12 catchment (m$^2$) to generate an estimate of local catchment flow (m$^3d^{-1}$). This annual mean flow was ultimately converted into units of m$^3s^{-1}$, the volume of water represented by the stream corresponding to that catchment. These calculations were performed at

individual level-12 catchments to estimate catchment-level mean annual flow (MAF, m$^3s^{-1}$). Aggregating HydroBASINS from all regions (excluding Greenland, parts of Arctic and the Antarctic), there were 1,017,091 level-12 catchments covering the globe at which MAF was estimated. For areas not covered by the MAR grid due to limitations of data availability, MAF was not estimated. The MAFs thus created represent discharge from each catchment as a discrete unit and do not account for flows from the hydrologic upstream contributions that make up for flows in a large stream or river.

**2.6 Hydrologic sequencing**

A simple and practical approach to catchment flow routing was utilized – flow from runoff is immediately routed through the catchments based on their hydrologic connectivity. After delineating individual level-12 catchment flow, cumulative flows (accounting for upstream flow contributions for each catchment) were computed by routing the hydrologic sequencing of the spatial catchment network and aggregating individual catchment flows. This was performed by utilizing the

Pfafstetter coding system within the HydroBASINS level-12 catchments (Lehner and Grill, 2013) to order the catchments hydrologically from upstream to downstream. For this purpose, a hydrologic sequencing program was created using the C programming language (Esri, 2021) which harnessed the coding system of the level-12 catchments for hydrologic routing. Documentation from the NHDPlus V2 dataset for the U.S. (McKay et al., 2012) was referenced for guidance in developing the hydrologic sequencing program. Hydrologic routing methods and channel storage, additional flows (e.g., subsurface

flows, snow melt, etc.), abstractions, or attenuation were not considered for refinements of the catchment flow.



## 2.7 River MAF estimation

Applying the hydrologic sequencing program, individual level-12 catchment MAFs were aggregated by hydrologically routing (upstream to downstream) the river network, including flows from all tributaries, generating an aggregated MAF across the river basin. This sequencing program was applied for individual river basin separately to cover the entire region

of HydroBASINS level-12 catchment dataset, combining MAFs to develop aggregated river flows for all river basins within the region. The process was then repeated for the 8-regions and aggregated MAF for all level-12 catchments (n = 1,017,091) covering the globe was estimated (Fig. 2(b)).

A spatial and tabular relationship (HydroRIVERS, 2019) exists between each HydroBASINS level-12 catchment and the corresponding river segments in HydroRIVERS which hydrologically represents the catchment. The river segments are

uniquely identified by the HYBAS_L12 attribute representing the level-12 catchment, however there are multiple segments per catchment indicating all rivers within a catchment. Since there was only one MAF calculated per level-12 catchment, all the river segments within the catchment represent the same MAF. To simplify this spatially, the multiple river segments per catchment were aggregated (using the 'dissolve' function in ArcGIS) to create a single HydroRIVERS river segment per level-12 catchment. This process was repeated for each region separately for the 8-regions thereby creating a total of

1,004,749 river segments covering the global extent.

The level-12 catchment and river segments are related through a tabular relationship with HYBAS_L12 as the common attribute, i.e., individual catchment can be joined to its corresponding river segment. Utilizing this relationship, the aggregated MAF calculated earlier for each level-12 catchment was transferred to the corresponding river segment in HydroRIVERS (using the 'join' function in ArcGIS), thereby providing MAF for the river network. This process was

performed for each region separately, developing the MAF for the entire river network covering the globe (n = 1,004,749 rivers) (Fig. 3). Aggregated MAF ($m^3s^{-1}$) representing flowing water at both level-12 catchments of HydroBASINS and the river network of HydroRIVERS was estimated at the global scale.



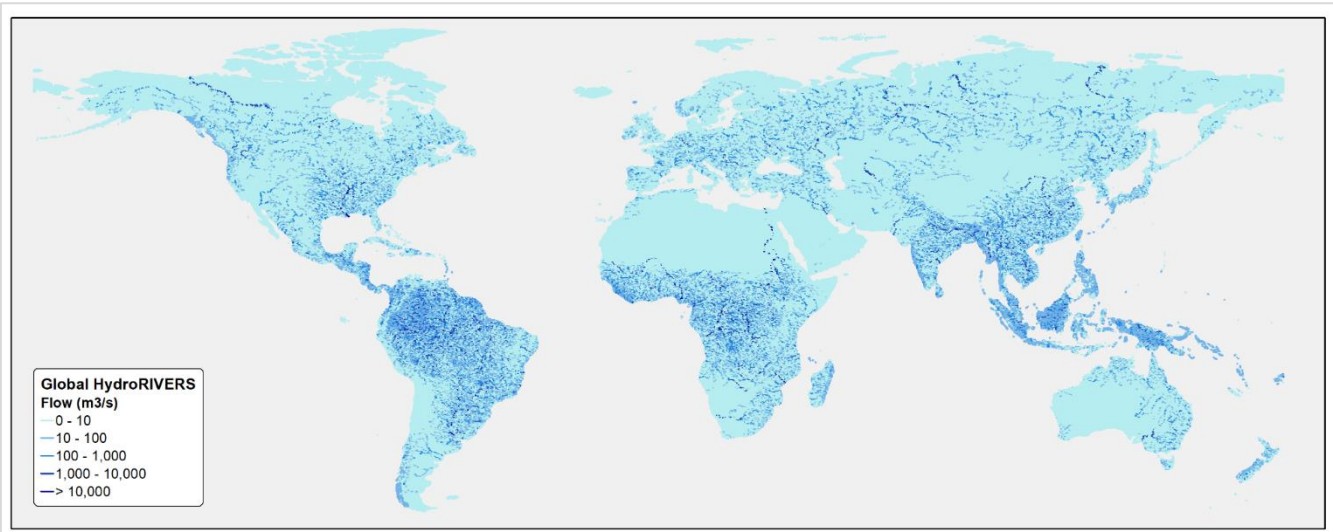

**Figure 3.** Estimated MAF ($m^3s^{-1}$) at the HydroRIVERS river network covering the globe

In most cases there exists a one-to-one relationship (HydroRIVERS, 2019) between level-12 catchments from HydroBASINS and river segments from HydroRIVERS, but there are instances where such relationship do not exist either due to inherent limitation of spatial resolution of source data (e.g., very small catchments) or due to other local hydrologic challenges like subsurface flows (e.g., springs or inland sinks with no visible surface outlet). Due to these limitations, total number of river segments in each region is slightly lower than the total number of level-12 catchments. At the global scale, there are about 1.2% (n = 12,342) fewer river segments with MAF data than the catchments.

Across the global river network, about 10.2% of the river segments (n= 102,674) resulted with an estimated MAF of 0 $m^3s^{-1}$. This was due to two distinct climatic conditions across parts of the globe: firstly, areas with insufficient precipitation and surface runoff resulting in non-existent river flows in certain arid and semi-arid regions of the globe (e.g., parts of North Africa, South-West parts of North America, Central Asia, most of Australia, etc.), and secondly, Arctic and Antarctic regions covered by ice where most of the precipitation is in the form of snow in winter and surface runoff is very challenging to quantify in such places (e.g., Northern Canada, parts of Northern Europe and Siberia, etc.). The river flows are not always high in areas with high precipitation as the flow is primarily dependent on the soil-land cover-precipitation mix across all parts of the entire river basin.

**Data Availability**

Two data spatial products from this project are available at https://doi.org/10.6084/m9.figshare.22694146 for public distribution (Heisler, et al., 2023). The first dataset is the global coverage of the aggregated MAF values on the HydroBASINS level-12 catchment levels, including the polygon level-12 catchments with the tabular attribute RIV_Q_CMS





representing MAF ( $m^3s^{-1}$). The second dataset is the global coverage of the aggregated MAF values on the HydroRIVERS
        river network, including the river line network with the tabular attribute RIV_Q_CMS representing MAF ($m^3s^{-1}$). Both
        datasets are spatial vectors distributed in the World Geodetic System 1984 (WGS84) datum geographic coordinate system
        for consistency with the source HydroBASINS and HydroRIVERS data.

**Code availability**

ArcGIS Pro version 2.1 of the ArcGIS suite of software was used for spatial data management and processing operations.
        GIS models were built within ArcGIS for spatial processing and used for repetitive calculations and operations with raster
        and vector data over the eight regions.

## 3 Results and Discussion

### 3.1 Evaluation of river flow

The resulting modelled river flow was evaluated by comparing to measured river flow at gauge stations. At the regional
        scale, it is challenging to compare the estimated river MAF to any measured flow since there is significant variability in the
        presence of measured flow at gauge stations by country, and public access to measured flow at these gauges is limited at a
        global scale. As such, in areas with readily available measured river flow data, quantitative comparisons were made to
        evaluate the estimated MAF by employing $R^2$ and RMSE (root mean square error) modelled values; where $R^2$ is a relative
measure of fit and RMSE is an absolute measure of fit in comparison to the range of estimated discharge data. Prior to these
        interpretations, all models were assessed to ensure relatively normal distribution of the model's residuals. Due to the
        complex nature of river discharge, all data points were retained unless determined to be an extreme outlier from the dataset.
        To compare data across the multiple geographies used, a comparison between models was also included by calculating and
        assessing each model's RMSE-Observations Standard Deviation Ratios (RSR) (Moriasi et al. 2007).

The NHDPlus V2 river flow dataset (USEPA, 2012) and Global Runoff Data Centre (GRDC, 2020) database were used
        for the measured gauge data comparisons. The NHDPlus V2 covers the entire United States, is publicly available, and is the
        most detailed large-scale hydrology with detailed catchments and river networks. While the global GRDC dataset is less
        detailed (i.e., fewer gauge locations per country), the U.S. only NHDPlus V2 data was used to demonstrate a regional
        comparison, using the Ohio River Basin as a case study, following previous iSTREEM® regional developments (Kapo et al.
2016, Wang et al. 2000). On a global scale, the GRDC dataset was used to make national-level comparisons. The
        comparison results (Table 4) are discussed in the following sections.





**Table 4.** Comparison of estimated river MAF ($m^3s^{-1}$) with gauge adjusted mean annual river flow from NHDPlus V2 for the Ohio River
and GRDC gauge measured mean annual discharge across 11 countries. *RSR = standard deviation ratios

| Country | $R^2$ | RMSE ($m^3s^{-1}$) | Estimated Discharge Ranges ($m^3s^{-1}$) | RSR* |
|---|---|---|---|---|
| Ohio River, United States (n = 167, 698) | 0.97 | 47 | 0 - 8,673 | 0.18 |
| United States (n = 936) | 0.90 | 522 | 0.01- 18,209 | 0.45 |
| Canada (n = 629) | 0.89 | 316 | 0.06 - 8,494 | 0.50 |
| Mexico (n = 60) | 0.91 | 196 | 1 - 1,942 | 0.31 |
| Brazil (n = 481) | 0.95 | 3,916 | 0.7 - 178,003 | 0.28 |
| China (n = 28) | 0.69 | 431 | 4 - 1,670 | 0.71 |
| Japan (n = 141) | 0.79 | 69 | 8 – 523 | 0.69 |
| India (n = 29) | 0.92 | 917 | 12 - 3,061 | 1.5 |
| Philippines (n = 46) | 0.87 | 24 | 0.5 – 258 | 0.36 |
| Germany (n = 334) | 0.98 | 74 | 0.1 - 2,321 | 0.23 |
| France (n = 297) | 0.76 | 75 | 0.1 - 1,723 | 0.48 |
| United Kingdom (n = 204) | 0.80 | 11 | 0.1 - 173 | 0.52 |

**Comparison with NHDPlus river flow data case study**

The NHDPlus V2 river flow dataset (USEPA, 2012) provides a robust and abundant flow data source for the U.S. NHDPlus
V2 river flow is based on runoff, temperature, precipitation and adjusted to USGS gauge flow measurements from 1971-
2000 (McCabe and Wolock, 2011). The Ohio River basin of about 422,000 $km^2$ with 167,698 gauge locations was used as
an example of a regional flow comparison.

Since NHDPlus catchments are much smaller in size than the level-12 catchments it was appropriate to create river flows
for NHDPlus catchments using the estimated MAR data. Utilizing the estimated global MAR (mm) dataset at 50m
resolution, MAFs ($m^3s^{-1}$) for each NHDPlus catchment and cumulative river flows (by hydrologic sequencing) across all
catchments within the Ohio River basin were estimated in the same manner as was done with the HydroBASINS level-12
catchments at the global scale. Estimated MAFs at NHDPlus catchments were compared with the corresponding gauge
adjusted mean annual flows from NHDPlus for the same catchments; just over 167,000 NHDPlus catchments were analyzed
for this comparison. Results showed a very strong correlation ($R^2$ = 0.97) (Fig. 4 (a)Error! Reference source not found.) with
the estimated MAFs being slightly lower than NHDPlus flows. The regression line (0.87) also indicated that the estimated
MAFs are in good agreement (within a factor of 1.1) but slightly lower than NHDPlus flows. A lower RMSE value of 46
$m^3s^{-1}$ (Table 4) shows good agreement when compared to the estimated flows to NHDPlus (Table 4). This process showed





that the CN based approach that uses local data on soils, land cover and precipitation to estimate MAR, and then MAF, provided confidence in applying the method to other geographies and with different sized catchments.




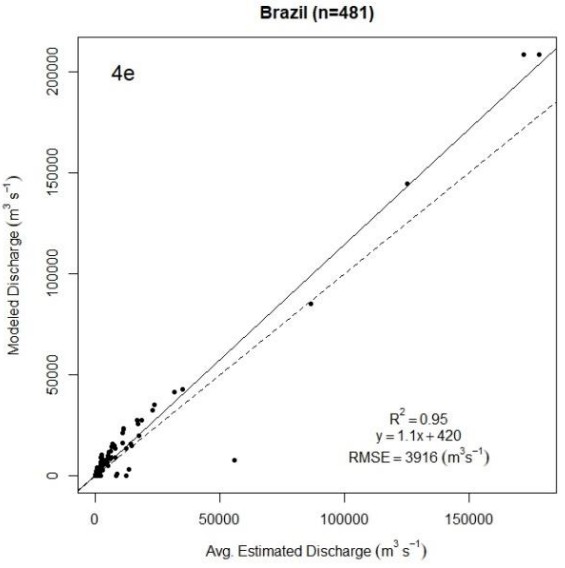

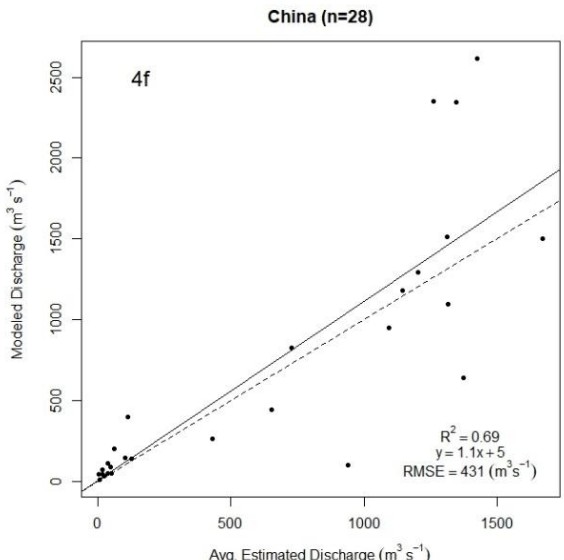


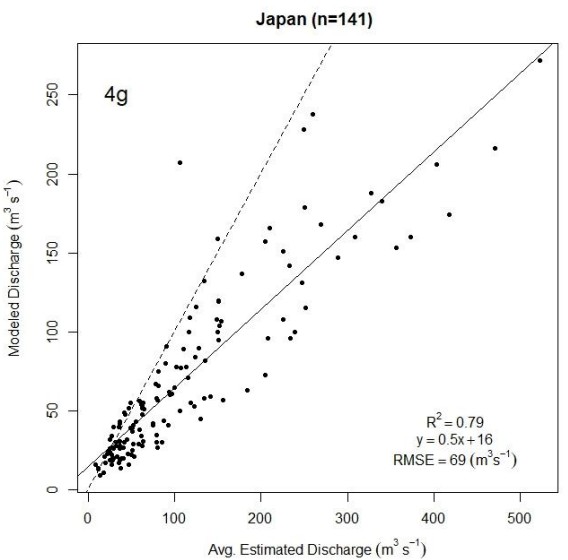

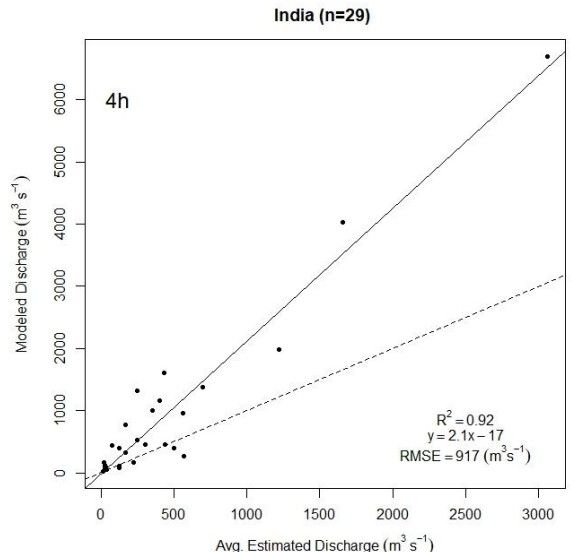




**Figure 4.** Comparison of estimated river MAF ($m^3s^{-1}$) with: (a) gauge adjusted mean annual river flow from NHDPlus V2 (n=167, 698) for the Ohio River; and GRDC gauge measured mean annual discharge in (b) United States (n=936), (c) Canada (n=629), (d) Mexico (n=60), (e) Brazil (n=481), (f) China (n=28), (g) Japan (n=141), (h) India (n=29), (i) Philippines (n=46), (j) Germany (n=334), (k) France (n=297), and (l) United Kingdom (n=204). The 1:1 agreement line is included on the plots as the dashed line.



### 3.3 Comparison with Global Runoff Data Centre gauge data

The Global Runoff Data Centre (GRDC, 2020) database provides daily and monthly observations of gauge stations monitored from the early 1800's, with variable record length. GRDC data were obtained for several countries in different

geographic regions across the globe to compare against the developed MAF of the rivers. Eleven countries were identified based on the priority for iSTREEM® expansion (McDonough, 2021), the impetus for this project, and to represent variance in geographic regions across the globe. Though the WorldClim 2 (Fick and Hijmans, 2017) precipitation data used for this project focused on a temporal range from 1970-2000, GRDC gauge data for that period for the countries of interest was limited in certain cases. For this exercise, from GRDC, about 4,000-gauges in the eleven countries along with spatial

locations and mean monthly discharge data ($m^3s^{-1}$) from the beginning of monitoring until 2020 were obtained. For each gauge location, mean monthly discharge for the years provided were averaged by year to arrive at a mean annual discharge ($m^3s^{-1}$) value for each gauge. Gauges in the eleven countries were monitored over a varied period of time, thus only those years and months that had data were averaged and used. Though the temporal ranges do not entirely overlap with the precipitation data representing 30-years (1970-2000) and the GRDC gauge data representing a varied time scale by country,

multiple years were considered to account for the variability from the dry and wet years over a larger period thereby aiding in better understanding the average flows.

    Some of the gauge location spatial coordinates provided by GRDC were inaccurate when compared to the gauge location descriptions provided. As such, these were manually checked and were moved to adjust to the gauge description provided. This inaccuracy could be due to the gauge location's latitude and longitude value limited to the first or second decimal (e.g.,

126.1, 34.88, etc.) which significantly affects the location accuracy. For example, when the gauge description was mentioned as 'station on Yellow River' in China and if it was miles away from the river, the gauge point was manually moved to the closest location on the river. This process was performed in ArcGIS guided by the aerial imagery data provided by Esri (2019). The number of gauges thus adjusted on a per-country basis is discussed below in the comparison. In certain cases, where the location could not be corrected with reasonable accuracy, these gauges were discarded from further use which

varied by country and resulted in 2.5% of total gauges (n = 95).

    In North America, the U.S., Canada, and Mexico were used for comparison. In the U.S., to conduct a nation-wide comparison, GRDC mean annual discharge data from gauge locations covering the entire U.S. was used to compare against estimated MAF at level-12 catchments where GRDC gauges spatially overlapped. The gauges in Hawaii were excluded due to lack of MAF data for that area. Across the continental U.S., the 1,014 gauges included those in Alaska that fall in the

Arctic zone, and this comparison showed good correlation ($R^2 = 0.82$). However, the comparison with flow gauges exclusively for Alaska showed moderate agreement ($R^2 = 0.55$). For gauges that fell in the Alaskan Arctic zone, estimated river MAF flows were consistently under predicted which indicated that flows from snowmelt and permafrost may not have been adequately captured by this method. When the gauges in Alaska were excluded and only those gauges in the continental U.S. were considered (n=936); the comparison showed a stronger correlation ($R^2 = 0.90$) (Fig. 4 (b)). The





regression line slope (1.21) indicates that the predicted MAF flows were also in good agreement, but slightly higher than the GRDC monitored values. An RMSE value of 522 m$^3$s$^{-1}$ (Table 4) indicates some variability between the estimated and measured flows but the model is still able to predict the data with relative accuracy.

The comparison of estimated MAF was performed against GRDC mean annual river discharge data for gauge locations in Canada, which also followed similar results as Canada also has Arctic zones. The gauge locations were distributed across the

country including the northern regions that fall in the Arctic zone. A comparison of flows at all the 1,153 GRDC gauge locations covering entire Canada showed moderate agreement (R$^2$ = 0.60). Similar to Alaska in the U.S., for gauges in the Arctic zone, estimated river MAF flows were consistently under-predicted also suggesting that flows from snowmelt and permafrost may not have been captured. When the gauges in the Arctic region (above the Arctic circle) were excluded, representing the gauges in the non-Arctic zone (n=629) (i.e., southern region of Canada below the Arctic circle), the

correlation improved (R$^2$ = 0.89) (Fig. 4 (c)). Regression showed that the estimated MAF flows were generally lower than the measured gauge when comparing the results to the 1:1 line (i.e., slope of 0.6) thereby underestimating the modelled discharges. This is likely due to underprediction of flow in countries that have a large area with high snowmelt and/or permafrost, even below the Artic zone. An RMSE of 315 m$^3$s$^{-1}$ (Table 4) also suggests an acceptable amount of error and shows overall good agreement of estimated flows with gauge flows in Canada.

For Mexico, the comparison of estimated MAF against river discharge at 60 GRDC gauges showed a strong correlation (R$^2$ = 0.91) (Fig. 4(d)) (five of the 60 GRDC gauges were spatially adjusted for location accuracy). The slope of the regression tracked just higher than above the 1:1 line (slope = 1.5) indicating that the estimated MAF values were slightly higher than the gauge measurements. An RMSE of 196 m$^3$s$^{-1}$ compared to the range of estimated discharges, ranging between ~ 1 and 2000 m$^3$s$^{-1}$, indicates some variability between the modelled and estimated flows in Mexico. Based on the

variation present, the model was still able to reasonably predict flows given the smaller amount of data as comparison, and an increase in publicly available data would provide additional confidence in the accuracy of the model.

In South America, estimated MAF in Brazil was compared to measured data from 481 GRDC gauges (Fig. 4(e)). The comparison showed a strong correlation between estimated MAF and GRDC gauge data (R$^2$ = 0.96) (20 GRDC gauges had to be spatially adjusted for location accuracy) and the regression line nearly aligned with the 1:1 line (slope = 1.1) indicating

a strong agreement between estimated MAF and gauge data. An RMSE of 3916 m$^3$s$^{-1}$ (Table 4), suggests that the model produces acceptable discharge predictions accounting for the large variability in Brazil estimated discharges (1 to 178,003 m$^3$s$^{-1}$), with model residuals trending closely to the model's line of best fit.

China, Japan, India, and Philippines were all included for flow comparison in Asia. GRDC gauge data for China was very limited (n=28) and estimated MAF compared with mean annual river discharges from GRDC showed a reasonable

correlation (R$^2$ = 0.69) (Fig. 4(f)) (7 gauge locations were corrected for spatial accuracy and one site was removed as an extreme outlier). These results suggest the variability in the modelled discharges can be attributed to the estimated GRDC discharged used for testing. The slope of the regression for the China model (slope = 1.1) indicates the estimated MAF was generally in good agreement with gauge measurements. The RMSE value of 431 m$^3$s$^{-1}$ (Table 4) indicates reasonable



agreement of measured and estimated flows, however there is lack of measured data for larger river discharge locations
which could account for some variability in flows. Based on the results of the China model comparison, even with limited
data, there was good agreement, and it is anticipated that model results comparison would improve with more river discharge
information, particularly from locations in additional watersheds, representing ranges of conditions within the country.
Additional sources of measured flow monitoring programs or data in China could not be identified as many were not
publicly available for use and thus could not be used in the comparative analysis.

Similar comparisons were made for Japan with 141 gauges and resulting in an $R^2 = 0.79$ (Fig. 4 (g)), India with 29 gauges
with an $R^2 = 0.92$ (Fig. 4 (h)) (four locations manually corrected and one removed as an extreme outlier after confirming
location), and the Philippines, with 46 gauges, with an $R^2 = 0.87$ (Fig. 4 (i)) (nine gauge locations corrected). For Japan, the
estimated MAF were generally lower than the reported gauge measurements (slope = 0.5), however still within a factor of 2
indicating reasonable agreement, and the relatively high $R^2$ (0.79) indicating that the model was able to capture flow
variability. For India, the MAF estimates were generally higher (slope = 2.1), which indicates reasonable agreement given
that limited data for is available for comparison and that it is a large and geographically diverse country. The Philippines
MAF predictions were generally in very close agreement to gauge measurements (slope = 1.0). RMSEs of 69 $m^3s^{-1}$ for
Japan, 917 $m^3s^{-1}$ for India and 24 $m^3s^{-1}$ for Philippines (Table 4) indicate good agreement of the estimated MAF with gauge
flows based on the range of estimated discharges per country (Table 4), in addition to all slopes being within a factor of 2.

Germany, France, and the United Kingdom (U.K.) were used for flow comparisons in Europe. The comparisons in
Germany at 334 GRDC gauges showed a very high correlation with $R^2 = 0.98$ (Fig. 4 (j)) (five locations corrected), and
France at 297 GRDC gauges showed a good correlation with $R^2 = 0.76$ (Fig. 4 (k)) (nine locations corrected) and the U.K. at
204 GRDC gauges also showed a good correlation with $R^2 = 0.80$ (Fig. 4 (l)) (six locations corrected and 23 number were
removed due to a lack of predicted flows). MAF predictions for Germany and France suggest good agreement with the gauge
495 measurements resulting in regression slopes close to one (slope = 0.8, slope = 0.73, respectively). MAF predictions for the
United Kingdom were also reasonably well predicted, generally lower than gauge measurements with a regression slope of
0.58. Assessment of the gauge data from the U.K., indicates a lack of higher discharge locations in the dataset, which may
account for the slight model underprediction when compared to other European countries. RMSEs of 74 $m^3s^{-1}$ for Germany,
11 $m^3s^{-1}$ for the U.K., and 75 $m^3s^{-1}$ for France (Table 4) all indicate good agreement of the estimated MAF with gauge flows,
with the three model's residuals trending closely to each line of best fit (Fig. 4 (j, k, l)). When compared to the range of
estimated discharges, all three models displayed an acceptable variation for conservative applications (Table 4), and
predicted flows were generally within a factor of two of gauge measurements indicating good model performance.

To compare model results from hydrologically diverse regions of the globe we calculated and compared RMSE-
Observations Standard Deviation Ratios (RSR) (Golmohammadi et al., 2014). RSR values vary from an optimal value of
0.00 to a large positive value, with lower values indicating higher model simulation performance (Golmohammadi, et al.,
2014; Moriasi, et al., 2007). In review of the literature, the RSR was selected to rely on more than validation metric and on
the recommendations of Singh et al. 2005 to incorporate a normalized error index. We applied a rating scale similar to the





one proposed by Singh et al. 2005 and applied by Moriasi et al. 2007; rating RSR values <0.50 as "Very Good" with a decrease in rating for each 10% increase in RSR value (Table 5).

**Table 5.** RSR model rating applied in our data assessment as proposed by Moriasi et al. (2007)

| Performance Rating | RSR |
|---|---|
| Very Good | $0.00 \leq 0.50$ |
| Good | $0.50 \leq 0.60$ |
| Satisfactory | $0.60 \leq 0.70$ |
| Unsatisfactory | $> 0.70$ |

Comparing model performance based on RSR values, the majority (8 out of 12) of evaluated regions had RSR values less than or equal to 0.5, the threshold for "very good", suggesting very good model performance in most cases (Fig.5). Additionally, two additional regions had RSR values <= 0.6 or =< 0.7, indicating these had "good" and "satisfactory"
performance, further indicating good to reasonable model performance across the regions evaluated. Of the regions evaluated, only two had RSR values above 0.7 (i.e., "unsatisfactory"), however, these were also the regions with least amount of gauge locations available for comparison (i.e. China, India). The amount of measured gauge locations available for each country was generally associated with better model performance (Table 4). Thus, it is likely that higher RSRs associated with these two countries is attributed to the limited gauge locations available for comparison, coupled with large
geographic areas and associated hydrological and meteorologic variability which may not necessarily capture the variability in flows across these regions.  As the model performance was "very good" to "satisfactory" for almost all locations (i.e. 10 out of 12), where there were more gauge locations for comparison, this suggests that the inclusion of larger datasets would substanstially improve the model performance evaluation. As noted above, we there are large datasets available for the countries selected, however, these are not publicly available, and our goal was to produce models using publicly available
information.

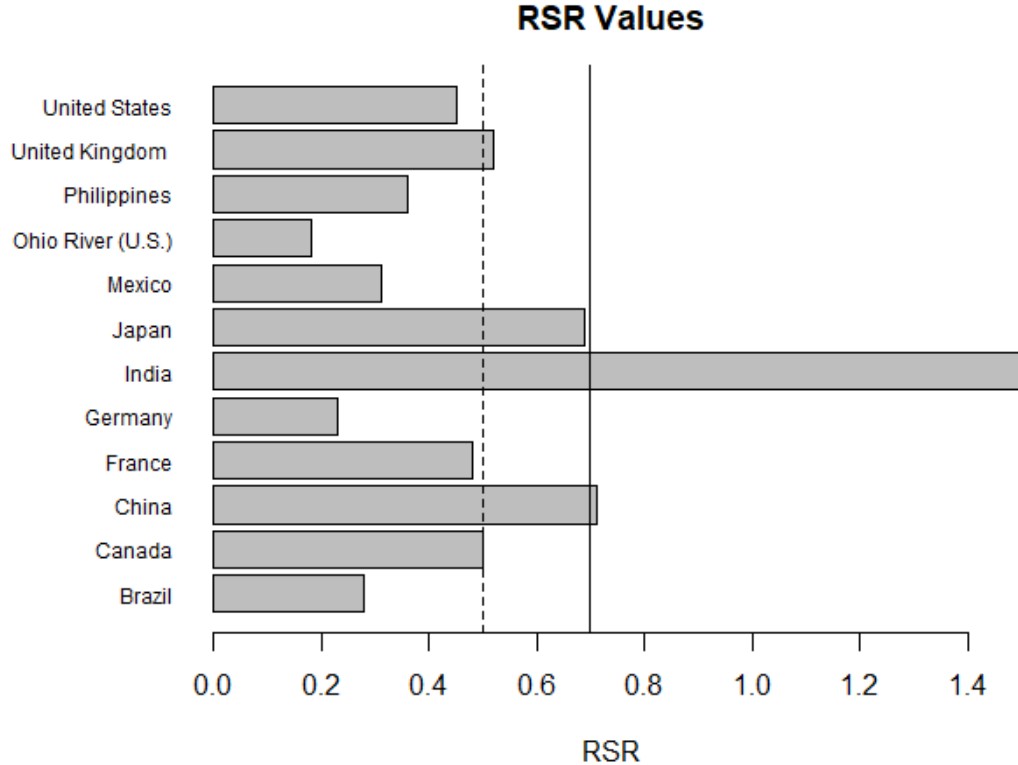

**Figure 5.** Comparison of RMSE-Observations Standard Deviation Ratios (RSR) for each country modelled. Following the assessment ratings as Moriasi et al. (2007), model performance was compared between counties to identify trends and limitation of each model and to determine the capabilities of the methodology we propose. Model Rating scale is provided in Table 5. All models with an RSR of < 0.5 we described as "Very Good" (*Dashed Line*) and models with RSR values >0.7 were rated as "Unsatisfactory" (*Solid Line*).

**4 Conclusion**

Across the twelve country comparisons between monitored and modeled data, the $R^2$ values from the linear regressions ranged from 0.69 – 0.98 indicating good overall correlation with measured data. Additionally, the slopes of the linear regressions were within a factor of 1 for six of the twelve countries, and within a factor of 1.5 for ten of the twelve countries, and within a factor of 2 for all twelve countries evaluated. Since the estimated MAF both over and under-estimated when comparing with gauge measurements, it suggests that there is not necessarily a bias in one direction, however, these differences could occur from variability in number of flow gauges available and gauge location accuracy, temporal range of flow measurements, and other location-specific factors such as high snowmelt or permafrost. The RMSEs across these comparisons ranged from 11 to 3,916 ($m^3s^{-1}$) indicating reasonable agreement between the estimated flows and measured

flows per country (Table 4). From these comparisons across diverse regions across the globe, it was found that MAF estimated based on the CN approach was effective in estimating flows across a wide selection of landscapes. As future work continues in modelling global surface waters, more granular gauge data will help with improving model evaluation. Higher resolution monitoring data, similar to that used in our Ohio River evaluation regional case study, would potentially better 545 capture variability across countries. The evaluations presented in this paper, however, provide reasonable confidence that these estimated MAF data can be used for applications at local and regional scales. The resulting spatially resolved global river flow dataset provides a useful tool that can be leveraged to support various scientific applications including for use in chemical safety assessments of down-the-drain chemicals across the globe (McDonough et al., 2021).

**Author contributions**

All authors equally contributed on this research project and reviewed the manuscript.

**Competing interests**

The authors declare no competing interests.

**Disclaimer**

**Acknowledgements**

The authors thank W. Marty Williams for brainstorming on approaches for surface runoff calculation and Brenna Kent (Waterborne Environmental, Inc.) for assistance with gauge flow comparisons.

**Financial Support**

The American Cleaning Institute and Procter and Gamble provided funding for this research.

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
