# Peer review of "Global river flow data developed from surface runoff based on the Curve Number method"

_Earth System Science Data, 2023_

## Author Comment (AC1)

**Combined responses to Reviewers' comments**

Vamshi et al. "*Global river flow data for aquatic exposure models developed from surface runoff based on the Curve Number method*"

ESSD essd-2023-161

**Reviewer 1**

*"Thank you very much for inviting me to review this manuscript. In this paper, the authors produce a dataset on global annual mean surface runoff and another dataset on global annual mean river flow rate. The topic is interesting and important, and the method used in this study have been explained detailedly. Nonetheless, in my opinion, the manuscript should be further improved before it can be finally accepted, and I am a bit worry about the novelty of this study."*

**Response:** We would like to thank the reviewer for the very helpful input provided that helped to improve the manuscript and clarify some key points in the previous manuscript version. Additionally, we have revised the introduction to elaborate on the focus of the key application and novelty of this dataset, and is discussed further below.

**Comment (1)**

(a) "*As resolutions of all input data (land cover, soil group & precipitation) for driving the CN approach is larger than 250 m, why the runoff is produced at 50 m? Why not produce runoff data at 250m, 100 m, or 10 m? In addition, for many large rivers, the width of the river channel can be hundreds or thousands of meters. A high resolution, like 50 m, is necessary only when you are going to calculate the water flow in small rivers. However, in your approach, the small rivers in each of the small level-12 catchment have been aggregated. For a global river flow database, is it necessary to produce river discharge data at a resolution of 50 m?*"

(b) "*To my understanding, the only outstanding point of the runoff and river flow datasets produced in this study, compared with previous datasets, is the high spatial resolution (50 m). However, as I have expressed above, a spatial resolution of 50 m is not necessary for larger rivers. It can be very useful to calculate the river flows in small rivers. However, the authors have aggregated all small rivers in each small catchment.*"

**Response:**

(a) Thank you for your question and we realize that this can be clarified in the manuscript. A common spatial scale was needed to aggregate the sources datasets of differing scales. That is, the resolutions ranged from 250m to 1 km (more specifically: 250m for the Soil Hydrologic Group (SHG); 300 m for the Land Cover; 1 km for the precipitation data). These were all spatially downscaled (using the resampling technique) from their native resolution to 50m as a common denominator for the spatial processing as part of this work. This step was necessary to avoid loss of information from the source datasets by using a lower spatial resolution and to create a common grid for aggregation of the geographic datasets. 50 m was selected as a reasonable common scale for processing the data sets. We have also updated the text of the abstract and Figure 1 to clarify that the spatial resolution of the final data products are on a level-12 catchment resolution.

We clarified in the manuscript why we scaled down the original datasets to 50 m and how the geographic datasets were reaggregated and processed. The following text was added to **Section 2.3** in the **Methods**:

"The hydrologic soil groups, land cover and newly created annual mean daily precipitation gridded datasets were resampled by applying the nearest neighbor method (using the 'resample' function in ArcGIS) to a 50 m grid. This step was necessary to create a common denominator to process the spatial data. The 50 m interval was selected as a reasonable spatial scale to aggregate and process the source data sets."
We also agree with the reviewer that the 50 m gridding is not necessary for a global river dataset as the 50 m resolution was used to aid in the spatial processing of the source data for the run-off but was not used for the final data products. The 50 m grid run-off data was used as input to the river flows, which were aggregated up to the Level-12 catchment resolution. We have added the figure below to the discussion in **Section 2.4** to help clarify these processing steps resulting in flow at Level-12 catchments. Additionally, we have clarified in the text that the resolution of the flows are at Level-12 catchments.

[Figure]

(b) We agree and appreciate this comment since it helped us to better address the focus of the work and the novelty of our work. We addressed the reasoning behind the 50 m data processing and catchment scale aggregation in A1. The intent of this work is not improving the flow resolution compared to previous datasets, but to provide a flow dataset suitable for exposure model applications. We agree with the reviewer that this was not clear in the previous version of the manuscript, therefore we have made clarifications in the text as follows:

- We modified the Title to "Global river flow data for aquatic exposure models developed from surface runoff based on the Curve Number method" to further elucidate the target of the model.
- We explained in the introduction why the CN method was selected to simulate river flow and its advantages in exposure model applications.
- We removed the wording "50 m resolution" in the manuscript, since this created confusion, and replaced with "50 m sampling" / "50 m gridding" in the methods section and focus the language on the spatial resolution of the final product of flows at Level-12 catchments.

**Comment (2):**
"*I would suggest the authors to further evaluate the interannual variation of the simulated river flows in this study using the observations. For such a dataset, it will be great if the scheme used in this study can capture the change trend of river flow. Otherwise, the datasets produced in this study seems to be not that useful*."

**Response:**

We appreciate the suggestion of the reviewer and indeed this could be an interesting exercise as a follow-up to this work. However, the intent of the current model was not to provide a historical dataset of river flow but build a predictive approach suitable for simulating mean yearly flow, which is the metric used environmental exposure models globally across regulatory frameworks (for example in the European Union). An evaluation of the interannual viability of the flow was therefore out of the scope of this work, which also includes the evaluation of the approach, and thus exploring seasonable variability using this approach would be a natural next step for a future study. We have clarified the intended use of this flow dataset in the Introduction.

Additionally, we further addressed this point in the manuscript by introducing an **Uncertainties and limitations** section where we discussed this point:
- **Section 4.3 Temporal resolution:** discusses this limitation of the current model and potential next steps to add in temporal variation
- **Section 4.5 Implications for risk assessment:** Notes this as potential follow up.

**Comment (3)**

*"Is it possible for the authors to compare the accuracy of river flows simulated in this study with existing global datasets of river flow? Are the results of this study more accurate than existing datasets?"*

**Response:**

We have added clarifying text explaining why we chose to compare to measured gauge data rather than other flow datasets, such as FlO1K. The comparison with measured gauge flow data from GRDC was considered to be an optimal benchmark to compare the mean average flow dataset at the gauge locations as these are measured rather than predicted data. It is difficult to compare to other flow datasets at locations where flow has not been measured since it is unknown how accurate the flow predictions are for ungauged river segments. There are uncertainties associated with validating against global flow datasets for ungauged rivers since "accuracy" on those ungauged segments cannot be verified. Each river flow dataset has a different approach to calculating flow and contains their own uncertainties.

Thus, we chose to only evaluate our data against measured gauge locations and have added a sentence to clarify this in the manuscript in **Section 2.8**:

*"Although flow datasets are available in the literature at global scale (for example, the aforementioned FLO1K model (Barbarossa et al. 2018)), the flows at ungauged sites are estimated rather than measured, and thus this could introduce bias into the model evaluation as comparison are made to modelled values. Thus, the point-to-point comparison with measured flow at the gauge locations was considered the most appropriate benchmark to assess the performances of the model."*

**Comment (4)**

*"I did not find any discussion on the uncertainties in the datasets produced in this study. The authors have calculated the surface runoff and river flow using the simple empirical CN equations, and the CN equations only consider land cover and soil type. However, many other factors, such as topography, dam/reservoir, irrigation, underground drainage and temperature, can also strongly affect river flow. I would suggest the authors to add some discussion on these factors."*

**Response:**

The authors agree that flow is variable and other factors will affect the river flow, and although based on empirical equations, the CN number method has the great advantage of requiring minimum input data, making it feasible for the application on a global scale. We agree that a discussion on limitations and uncertainties would strengthen the paper and we have added an "**Uncertainty and Limitations**" section (**Section 4**) in the revised version of the manuscript that addresses these limitations. This section addresses

We have clarified the advantages of using the CN approach in the paper in in the **Introduction**, added discussion on processes that are not captured in the CN approach in the **Uncertainties and Limitation Section 4.4**, and added discussion on the implications of these assumptions and the applicability of the data for use in ecological risk assessment in **Section 4.5**.

**Comment (5)**
"*In Figure 1, the authors have provided a nice flowchart to show the approach for calculating surface runoff. I would suggest the authors to add a detailed flowchart to show the approach for calculating the river flow based on the surface runoff, HydroBASINS and HydroRIVERS.*"

**Response:**
We agree that a flow chart will help clarify the workflow. The flowchart below has been added to the manuscript.

[Figure]

**Minor Comments**
**(1)** "*L15 ** to create a global gridded dataset of annual mean surface runoff at a spatial resolution of 50 meters.*"
**Response:** We have amended the sentence and the new sentence reads: "to create a global dataset of annual mean surface runoff."

**(2)** "*L160: how the monthly precipitation data is converted into daily precipitation rates? Did you assume the precipitation is evenly distributed in each month?*"

**Response:** We have added text to the manuscript to clarify how the daily precipitation rates were derived, as this indeed was not clear in the original text.
The following text has been added to the **Section 2.1.3 Precipitation:**

"As the CN approach to estimating runoff relies on daily precipitation, the monthly rainfall values were converted to a single daily rainfall, representing the average precipitation per day over the year, for use in Equation (2). To do this, a yearly average rainfall per month was calculated by summing the amount of precipitation per month (i.e. across 12 months) and then dividing by 12; which was then divided by 30 days to arrive at an average amount of precipitation per day."

**(3)** "*L225-280: Based on the description in sections 2.5, 2.6, surface runoff has been calculated for each day using Eq. 2. Why not produce the river flow rate for each day or each month?*"

**Response:** We used this approach as we were modeling for an average daily amount such that the precipitation within the runoff grids did not vary over time. Therefore, the surface runoff (and as a result, the flow) was not calculated for each day and thus did not vary over time. We recognize that this is not the traditional approach to derive the runoff with the CN method, since the CN equation has been developed using daily data.
We have also added in discussion of this approach in the uncertainty section.
Clarification text was added to the manuscript as follows in **Section 2.1.3 Precipitation**:
"*As the goal was to produce average precipitation per day over the year, the precipitation within each grid did not vary over time using this approach. While rainfall intensity and total amounts of precipitation can vary daily, the purpose was to compute average total daily precipitation rates (inches per day) to predict average daily runoff in Equation. (2). As this is a simplifying assumption in the application of the CN approach, it may introduce some limitations in the flow predictions which are discussed further in Section 4. As part of future work, potential refinements can be implemented for specific geographies if spatially resolved rainfall data at daily scale are available.*"

**(4)** "*Fig. 2: the unit of surface runoff in panel (a) should be mm d-1, and the unit of water flow in panel (b) should be changed to m3 s-1*"

**Response:** The units have been corrected and were changed on the figures in the revised manuscript.

**Reviewer 2**

**Overview**: "*The paper describes the development of global dataset of mean annual flow obtained by using the Curve Number method.*"

**General Comments**

*"The paper is fairly well written and well structured. The topic is of interest for the readership of ESSD as a 50-m resolution dataset of mean annual flow (MAF) could have high impact in the hydrological community.*

*I read the paper with interest but I have contrasting feeling. The dataset, mainly due to its very high spatial resolution, might be relevant, but the method used for its development has several issues that must be carefully addressed for having the paper published. I believe that the actual spatial resolution is much coarser and that the usability of the datasets in several areas is limited.*

*To clarify, I have listed below the major issues with the indication of their relevance.*"

**Response:**

We would like to thank the reviewer for the very helpful input provided that helped to improve the manuscript and clarify some key points in the previous manuscript version. We agree with the reviewer that the resolution of the final dataset is coarser than 50 m and is of Level-12 catchment resolution and we have clarified this in manuscript. Based on your helpful comments, we have also clarified the primary goal and purpose of this work is for use in creating a flow dataset suitable for exposure model applications. In order to clarify these aspects in the paper:

- We modified the title of the manuscript to indicate the primary application is environmental exposure modeling as follows: "Global river flow data for aquatic exposure models developed from surface runoff based on the Curve Number method"
- We added additional context in the Introduction on the primary application of the dataset is for aquatic exposure assessment
- We removed the wording "50 m resolution" in the manuscript, replaced with "50 m sampling" / "50 m gridding" and clarified that the final flow data product is on the Level-12 catchment level and that the 50 m runoff grid was used as a processing step (we have provided additional details on this is the response to the next comment)
- We have added an Uncertainty and Limitations section to discuss the assumptions and limitations of the methods so that it is clearer how to appropriately use this data. Several of these aspects are discussed in the responses in the comments below, as well as in responses to Reviewer 1.

**MAJOR:** "*The main problem is related to the spatial resolution, or better sampling, of the developed dataset. The dataset is distributed at 50 m resolution but the forcing, i.e., precipitation, and mainly the method is not appropriate at such resolution. The method does not consider human impact on streamflow, such as reservoirs, wetlands, water diversions for agricultural, civil and industrial water uses. Other missing processes are related to high altitude regions, such as snow melting, glacier processes. If we want to deliver a MAF dataset at 50 m resolution, these processes must be included. Otherwise, in many areas, the MAF dataset here developed has very little reliability and applicability. This must be discussed, and better it should be fixed.*"

**Response:**

Thank you for your question on the spatial resolution and we see that this can be clarified in the manuscript. The 50 m spatial scale was used as a processing step for the runoff data, and then these were further processed into level-12 catchment flows. We realize that this was unclear in the manuscript. We have clarified this by removing references to 50 m resolution throughout the paper (and replacing with "50 m sampling" / "50 m gridding"), indicating that the final flow data product is on a level-12 catchment resolution in the abstract, and have revised Figure 1 to reflect this.

The revised text in the **Abstract** is as follows:

"These runoff data were then spatially combined with publicly available global hydrological datasets of catchments and rivers to estimate daily mean annual flow (MAF) across the globe on a level-12 catchment scale."

We used 50 m sampling for the runoff as a common spatial scale to aggregate the source datasets of differing scales. That is, the resolutions ranged from 250m to 1 km (more specifically: 250m for the Soil Hydrologic Group (SHG); 300 m for the Land Cover; 1 km for the precipitation data). These were all spatially downscaled (using the resampling technique) from their native resolution to 50m as a common denominator for the spatial processing as part of this work. This step was necessary to avoid loss of information from the source datasets by using a lower spatial resolution and to create a common grid for aggregation of the geographic datasets. 50 m was selected as a reasonable common scale for processing the data sets.

We clarified in the manuscript why we scaled down the original datasets to 50 m and how the geographic datasets were reaggregated and processed. The following text was added to **Section 2.3** in the **Methods**:

"The hydrologic soil groups, land cover and newly created annual mean daily precipitation gridded datasets were resampled by applying the nearest neighbor method (using the 'resample' function in ArcGIS) to a 50 m grid. This step was necessary to create a common denominator to process the spatial data. The 50 m interval was selected as a reasonable spatial scale to aggregate and process the source data sets."

We have also added the figure below to the discussion in **Section 2.4** to help clarify these processing steps resulting in flow at Level-12 catchments.

[Figure]

We agree with the author that the human impact of streamflow and other missing processes could be included in the dataset to improve flow predictions. While the CN approach is based on empirical equations, it has the great advantage of requiring minimum input data, making it feasible for the application on a global

scale, which is what we tested in this work. We addressed this by adding an Uncertainty and Limitations section in the revised manuscript mentioning these limitations and items that may be considered for future research. Specific examples where these limitations can impact prediction accuracy were reported as well.

**MAJOR:**

**(a)** *"A second major problem is related to the choices made for the application of the method. The method is applied by using monthly precipitation data disaggregated (simply the monthly values divided by the number of days) at daily resolution and it is not appropriate. Several global scale datasets of precipitation are currently available. The choice of the 1 km climatological dataset is questionable."*

**(b)** *"CN values for intermediate antecedent moisture conditions seem to be used, and it has no sense in many areas of the world."*

**(c)** *"It is not clear how the time of concentration at basin scale is used, and similarly the travel time for the river"*

**(d)** *"Actually, I believe that the way the method was used in the paper to develop the dataset is not appropriate. It must be clarified and discussed in details."*

**Responses:**

**(a)**

We agree that there are other global precipitation datasets are available, however WorldClim was used as it represents an appropriate choice currently available for modeling global mean annual flow with the CN method. Other datasets at daily temporal scale have lower spatial resolution or they rely on predicted rather than measured data. We chose the WorldClim (Fick and Hijmans, 2017) because 30 years (1970 – 2000) of global daily measured weather data were already compiled from seven different sources and the data were aggregated into monthly climate averages. Additionally, this dataset had been validated by Fick and Hijmans (2017). WorldClim is a global dataset that was developed by spatially interpolating weather station data of monthly average total precipitation, and it is available at a resolution of 1km.

This reasoning behind this choice was clarified in the manuscript as follows in **Section 2.1.3 Precipitation:** "Global datasets with daily temporal resolution are available in the literature, but the spatial resolution is generally on a larger scale, i.e. 0.5° lat. x 0.5° long. (~ 55 km), for example, the CPC Global Unified Gauge-Based Analysis of Daily Precipitation (Xie et al. 2007; Chen et al. 2008), or they are based on rainfall predictions rather than measurements, for example Karger et al. (2021). As the intent of this work was to develop a spatially resolved flow dataset for exposure models, higher spatial resolution was prioritized over temporal resolution and long-term measured rainfall data were considered more reliable than estimated values."

We disaggregated monthly data into daily data in order to apply the CN method. This is a simplifying assumption used in the application of the CN method and as such we have added into the main text a discussion of the limitations of this approach.

To clarify this (and potential implications), we added this text in the manuscript in **Section 2.1.3 Precipitation:**

"As the goal was to produce average precipitation per day over the year, the precipitation within each grid did not vary over time using this approach. While rainfall intensity and total amounts of precipitation can vary daily, the purpose was to compute average total daily precipitation rates (inches per day) to predict average daily runoff in Equation (2). As this is a simplifying assumption in the application of the CN approach, it may introduce some limitations in the flow predictions which are discussed further in Section 4. As part of future work, potential refinements can be implemented for specific geographies if spatially resolved rainfall data at daily scale are available."

We agree that this simplification may introduce some limitations in the predictions, and we discussed this by introducing an "Uncertainty and Limitations" section (**Section 4**). More specifically, the authors acknowledge that rainfall is variable over time in amounts and intensity. There are days with no rainfall and days with extreme rainfall events. Using the average rainfall event does not capture the extreme precipitation events that would create more runoff, and on the other hand, if there are droughts then the concentrations may be underestimated. This discussion was added to the manuscript in **Section 4**.

Furthermore, for the purposes of this study, (the goal of this program is to develop a flow dataset for spatially resolved aquatic exposure models for chemicals for use in environmental risk assessment across large geographic regions), it was considered fit-for-purpose.

In the Introduction, we have clarified the primary purpose of this dataset is for use in aquatic exposure models and also changed the title of the manuscript so that it is clearer to the audience the purpose and limitations of this dataset.

**(b)**

We agree that antecedent moisture condition II (AMC II) may not be appropriate for certain areas (i.e. desert areas, glaciers) and can vary over time even in the same area. However, the AMC II was considered most appropriate given the need for average flow conditions, for the purpose of developing a mean annual flow dataset for ecological risk assessments. We have addressed this in the manuscript by adding the following text to **Section 4:**

"As stated in Section 2.3, AMC II (i.e., representing average moisture conditions) was used for the CNs computation which was considered the most appropriate to simulate average flow conditions. It should be noted however that AMC can fluctuate depending on moisture conditions with AMC I representing dry conditions and AMC III representing wet conditions on a finer temporal scale. AMC can vary daily depending on previous rainfall events. By using AMC II the variability of the soil conditions, i.e. elongated dry periods, wet seasons, is was not accounted for. Future refinements to this framework may include a differential representation per climatic areas on a finer scale, i.e. arid regions may be better represented with AMC I curve numbers and tropical regions may be better represented by using AMC III curve numbers when modeling seasonal or flows on a finer temporal resolution."

**(c)**

Time of concentration at level-12 catchment scale was not taken into account and we have added to text to clarify reasoning for this the Uncertainties and limitations Section. We chose to not use time of concentration as it is not relevant when modeling on a yearly average basis and this is a common assumption used when modeling on a yearly time scale, for example, in well-established pesticide run-off models, for example, FOCUS SWASH and USA EPA Pesticide in Water Calculator.

Pesticide in Water Calculator (USEPA, 2020), and the FOCUS Surface Water used in the SWASH (Surface WAter Scenarios Help) model  (FOCUS 2003; FOCUS 2013)

Clarifying text has been added to the manuscript in **Section 4.3 Temporal resolution:**

"It was assumed that all the surface runoff in the catchment reaches the river segment at the same time, i.e. the time of concentration within the catchment was not considered, since this is not considered relevant when modelling on an average yearly basis. This is a common assumption used when modeling on a yearly time scale, for example, in well-established pesticide run-off models, such as FOCUS SWASH (FOCUS 2003; FOCUS 2013) and the USA EPA Pesticide in Water Calculator (USEPA, 2020). Potential refinements can be implemented within this framework for specific geographies if spatially resolved rainfall data at daily/hourly scale are available."

**(d)**

We have substantially modified and added to the manuscript to clarify all the assumptions made, the reasoning behind it, their limitations, and the potential impact for the intended use on the model applicability, as discussed in responses to comments from both reviewers.

**MODERATE**: "*The developed dataset is compared with data over US, and GRDC data globally. Currently, 30-year reanalysis datasets developed everywhere in the world are available. Such datasets should be used for the assessment of the developed dataset*."

**Response:**

We have added clarifying text explaining why we chose to compare to measured gauge data rather than other flow datasets, such as FlO1K. The comparison with measured gauge flow data from GRDC was considered to be an optimal benchmark to compare the mean average flow dataset at the gauge locations as these are measured rather than predicted data. It is difficult to compare to other flow datasets at locations where flow has not been measured since it is unknown how accurate the flow predictions are for ungauged river segments. There are uncertainties associated with validating against global flow datasets for ungauged rivers since "accuracy" on those ungauged segments cannot be verified. Each river flow dataset has a different approach to calculating flow and contains their own uncertainties.

We have clarified in the text why we chose to use the comparison with the gauge measured data was the most appropriate approach for this assessment in **Section 2.8:**

"*Although flow datasets are available in the literature at global scale (for example, the aforementioned FLO1K model (Barbarossa et al. 2018)), the flows at ungauged sites are estimated rather than measured, and thus this could introduce bias into the model evaluation as comparison are made to modelled values. Thus, the point-to-point comparison with measured flow at the gauge locations was considered the most appropriate benchmark to assess the performances of the model.*"

**MAJOR: "***The high values of R-square obtained in the paper are mostly related to the difference in river discharge from basins of different size. Larger basins have larger river discharge than smaller ones. A more robust assessment should be carried out by computing the mean annual flow normalized by the basin area. Moreover, section 3.3 should be strongly summarized. The assessment in terms of RMSE has little assessment for basins of*

*different size. Normalized scores should be used (as RSR, but its formula should be given). The gauges used for the comparison should be shown in a map. This section should be significantly revised.*"

**Response:**

Thank you very much for this very helpful comment. We have substantially added to the analysis of model comparison to measured gauge flow and made several revisions to this section. We believe that this analysis and section has been greatly improved and provides additional context and information on the performance of the model. These additions and revisions are summarized below:

- We agree that indeed the large range in flow (over orders of magnitude) have the potential to skew the correlations when comparing flows on a catchment level. To address this, we have added in analyses on log transformed flows which reduces the spread in the data and provides correlations that are less skewed toward low- or high-end flows. Each country comparison was also conducted on log transformed flows, with figures added to the **Appendix** (**Figure A2**) and discussed in the main text in Section 3.3. As such, we have also discussed the RMSE in terms of the log transformed flows which provide insight into the mean errors in terms of factors of agreement and have also added this to the discussion and the summary in Table 4. We elected to keep the flow comparison as well, as they do provide insights into agreements between modeled and measured flows in at specific locations on a catchment level, but these are juxtaposed with the log transformed comparisons in the discussion and **Table 4**.
- We have added in maps of the gauge locations to the **Appendix** (**Figure A1**) to clarify that we are making comparisons on the same catchments.

- As suggested, we also added in a comparison between measured and modeled flows on a basin level. We did this by summing flows across the GRDC sub-basin for each gauge station for each of the 11 countries. We also assessed this comparison on log transformed flows to account for range in flow values and calculated RMSEs resulting in a strong correlation of R2=0.9 and mean error of a factor of 2. Also on a sub-basin level, we calculated ratios of modeled to measured flows and plotted these for each sub-basin by country. These plots have been added to the main text in newly added **Figure 6** (see below). This analysis indicated that the majority of the sub-basin modeled flows fall within a factor of 3 of measured flow. The method and results/discussion of the analysis have been added in the manuscript in **Sections 2.8**, and **4.1**.

- We have substantially revised and summarized Section 3.3. We have moved a large part of the text describing the methods used in the comparison to a newly created subsection in the Methods as **Section 2.8 Model Evaluation**; we have revised Section 3.3 to include the analyses on the log transformed flows and summarized the results more concisely, and we added in a newly created section on the model uncertainty in terms of the flow comparison on a basin level, **Section 4.1 Model accuracy**.

    Newly added **Figure 6** comparing flows on a sub-basin level:

[Figure]

**SPECIFIC COMMENTS**

**(a)** "*L69: River discharge is generated by surface, subsurface and groundwater runoff, all components should be considered to obtain a physically reasonable approach.*"

**Response:**

We agree and have added an Uncertainties and Limitations Section that also mentions possible future research for refinement as follows in

**Section 4.4 Other processes contributing to the river flow:**
"The modelled flow dataset is the sum of mean annual runoff from all contributing areas along each river segment in a drainage basin. Contributions and losses not addressed in the calculations include subsurface return flow, groundwater recharge and discharge, extraction for irrigation and other anthropogenic activities such as flow augmentation from impoundments, or reservoirs. This level of detail is generally lacking at the global scale and these processes were thus not implemented."

**4.5 Implications for risk assessment:**
 "To simulate more realistic scenarios and better account for the temporal and spatial variability of the river flows, future refinements could include the development of seasonal conditions (i.e. performing simulations with different daily rainfall averaged for each month of the year and/or for each day of the year), CN slope-adjustments, addition of snowmelt runoff, and the inclusion of adjustments for subsurface runoff and groundwater contributions, diversions, and in-basin storage."

**(b)** "*L211-221: The comparison with GCN250 dataset is not clear. The comparison of maps with the computation of BIAS or RMSE should be carried out. It would be clearer.*"

**Response:**

We agree that this comparison and its purpose should be clarified. As such, we have added additional discussion on this comparison and moved it into the Uncertainty and Limitations section of the manuscript in **Section 4.2**:

"The source datasets used to implement the runoff calculation had a spatial resolution ranging from 250 m to 1 km, while the river flows were reaggregated to a level-12 catchment scale. These geographic datasets were selected as providing the best resolution currently freely and publicly available in literature at global scale for all input parameters. However, to allow for an efficient data processing, some assumptions were introduced such as the land use class aggregation discussed in 2.3. In order to validate this assumption, the GCN250 dataset (Jaafar, et al., 2019) of global gridded CNs was used for comparison.  The GCN250 dataset used the same soil dataset (i.e, HYSOGs250m (Ross et al. 2018), but a different underlying land use dataset (i.e. ESA CCI-LCESI (2018)) with more land use classes, and thus provided an appropriate dataset to evaluate the land use (and thereby CN) aggregations used in this study.  To do this, the CNs for soil group C (which was found to be the most predominant soil group across the globe, Table 3) from this study were compared to those in GCN250.  For each of the six land use types in this study (Table 2), the CNs from GCN250 set of the relevant land use types were averaged and then compared to the aggregated CN in this study.  For soil group C, the average CNs from the GCN250 dataset were 75, 74, 82, Not applicable/0, 88, and 92 for 'Cultivated land', 'Wood/forest land', 'Meadow', 'Open water', 'Urban', and 'Desert/bare' land types, respectively; and in the same order, from this study the CNs were 83, 74, 71, 0, 88, and 85.  Additionally, the small variation in CN for forest types in our aggregation was also found in the GCN250 dataset where the range in CN tree plant functional types (PFT) was relatively small, for example for soil type C, CN ranged from 70-77 across the tree PFTs.  The results from this comparison indicate good agreement and that the land use type aggregation used in our study were reasonable, appropriate, and fit-for-purpose."

**(c)** *"In Figure 2a it is hardly possible to distinguish the different classes. To be improved."*

**Response:** We have improved Figure 2 (a) to address this. The new figure is below

[Figure]

**(d)** *L384: typo*

**Response**: We have corrected this typo and the new sentence now reads "Results showed a very strong correlation ($R^2$ = 0.97) (Fig. 5 (a)) with the estimated MAFs being slightly lower than NHDPlus flows."

**(e)** *"L428: What is the number of adjusted gauges? It should be reported to GRDC to correct the database."*

**Response:**  We have clarified the text such that the number of adjusted gauges are listed all together. The following text was added to the manuscript in Section 3.3:

"*In summary the number of gauges spatially adjusted for each country were as follows 5 gauges in Mexico, 20 in Brazil, 7 in China, 4 in India, 9 in Philippines, 5 in Germany, 9 in France, and 6 in the UK*."

We did not report the spatially adjusted gauges to GRDC as GRDC reports a limitation that the gauge locations may not be accurate.

**(f)** *L523 typo.*

This sentence has been removed from the manuscript.